# A Prototypical Network-Based Approach for Low-Resource Font Typeface Feature Extraction and Utilization

**Kangying Li** [1,*] , **Biligsaikhan Batjargal** [2] and **Akira Maeda** [3]

1. Graduate School of Information Science and Engineering, Ritsumeikan University, Shiga 525-8577, Japan
2. Kinugasa Research Organization, Ritsumeikan University, Kyoto 603-8577, Japan; biligsaikhan@gmail.com
3. College of Information Science and Engineering, Ritsumeikan University, Shiga 525-8577, Japan; amaeda@is.ritsumei.ac.jp
* Correspondence: gr0319ss@ed.ritsumei.ac.jp

**Abstract:** This paper introduces a framework for retrieving low-resource font typeface databases by handwritten input. A new deep learning model structure based on metric learning is proposed to extract the features of a character typeface and predict the category of handwritten input queries. Rather than using sufficient training data, we aim to utilize ancient character font typefaces with only one sample per category. Our research aims to achieve decent retrieval performances over more than 600 categories of handwritten characters automatically. We consider utilizing generic handcrafted features to train a model to help the voting classifier make the final prediction. The proposed method is implemented on the 'Shirakawa font oracle bone script' dataset as an isolated ancient-character-recognition system based on free ordering and connective strokes. We evaluate the proposed model on several standard character and symbol datasets. The experimental results showed that the proposed method provides good performance in extracting the features of symbols or characters' font images necessary to perform further retrieval tasks. The demo system has been released, and it requires only one sample for each character to predict the user input. The extracted features have a better effect in finding the highest-ranked relevant item in retrieval tasks and can also be utilized in various technical frameworks for ancient character recognition and can be applied to educational application development.

**Keywords:** feature fusion; metric learning; low-resource data

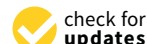



## 1. Introduction

The Greek philosopher Aristotle explained things, concepts, and symbols in his 'On Interpretation'. He emphasized the writing system and the complicated relationship among things, concepts, languages, and cultures. There are different ancient characters in the world. For example, the ancient Egyptian hieroglyphs lost their meaning in the fourth century AD and became a mysterious writing system. Protecting existing records of language and writing systems has become the goal for humanities studies. If these scarce records can be archived, the system that provides retrieval or identification can be accessed publicly. More scholars can see clearer vectorized glyph resources, and more people will be able to taste these fading historical–cultural ambiances.

Font typeface resources may be a good way to protect the ancient character records; for example, many ancient character fonts are made strictly based on ancient characters or symbols that are no longer used [1–4].

However, some of the ancient characters are represented by historians and archived with just one sample for each character. Although many character-recognition techniques have been developed, while most of them have been created for a specific domain, and each category of characters has sufficient training data [5–7], determining how to use the few sample resources effectively is still a challenge.

This research provides a perspective on the flexible use of low-resource ancient character datasets and tries to evaluate the performance of our proposed method on available datasets and tasks. The goal is to obtain features with better generalization capabilities from font typeface data and apply them to character-retrieval tasks.

### 1.1. Shirakawa Font

The Shirakawa Shizuka Institute of East Asian Characters and Culture, Ritsumeikan University, has developed a search system for the 'Shirakawa font' [2], which can be used freely by converting kanji characters from the modern calligraphic style to the old calligraphic scripts (oracle bone script, bronze script, and seal script). The total number of oracle bone script characters recorded was 681. Each ancient character was made as a vector image with a smooth edge, and users can enlarge each search result to see the details. These font typeface images have also been bundled as font packages and released on the website. Our research considers the efficient use of font typeface image resources and implements an oracle bone script offline handwriting-recognition framework based on the oracle bone script in the 'Shirakawa font'.

## 2. Related Work and State-of-the-Art

In the following sections, we discuss character-recognition techniques applied to ancient character datasets and compare their differences with the content-based image retrieval based on shape-matching.

### 2.1. Ancient Character Recognition

The digitization of ancient manuscripts and inscriptions is a very complicated and difficult process [8]. Many efforts have been devoted to overcoming resource scarcity and deformation caused by handwriting. As shown in Table 1, many techniques have been proposed and applied to different ancient characters. Some of them focus on the character characteristics of the object domain, extract some geometric features, and obtain good results on a target dataset.

**Table 1.** Related work on ancient character recognition.

| Script | Method | Task and Data Availability |
| --- | --- | --- |
| Devanagari script | Naive Bayes, RBF-SVM, and decision tree using HOG and DCT features [5] | 33 classes and 5484 characters for training; dataset is unavailable |
| Tamil character | Fuzzy median filter for noise removal, a neural network including 3 layers [9] | Total class number is unknown; dataset is unavailable |
| Batak script | K-Nearest Neighbors [10] | Total class number is unknown; dataset is unavailable |
| Vattezhuthu character | Image Zoning [6] | 237 classes and 5000 characters for training; dataset is unavailable |
| Odia numbers | LSTM [7] | 10 classes and 5166 characters for training; dataset is unavailable |

Unlike the previous research on ancient character recognition shown in Table 1, the main challenge of our work is that the total number of characters to be classified is 681 characters, and there is only one reference sample for each character. We aim to find a general method that uses public and sufficient data resources from other domains to perform the same retrieval task for public ancient character scripts.

### 2.2. Sketch-Based Image Retrieval Based on Shape-Matching

Many content-based image retrieval approaches use different representations of images, such as color, object shape, texture, or a combination of them [11]. A shape is one of the most common and determinant low-level visual features, which contains an object's

geometric structural characteristic. Sometimes, we could assume a character as a symbol with a shape.

There are two main classes of sketch-based shape matching approaches: (1) the learning-free method and (2) the learning-based method. As with traditional image retrieval methods, learning-free methods focus on selecting representation, extracting global [12] or local [13] features, and accomplishing retrieval based on a nearest-neighbor search in a descriptor space. The learning-based method [14] usually requires sufficient annotated data. It tends to work better than learning-free methods but usually involves massive manual-annotation effort. Shape-matching can usually transform a shape and measure the resemblance with another using some similarity metrics. The problem is how to evaluate the degree of similarity. Some feature descriptors do not pay much attention to gradient orientation in localized portions of an image, for example, features extracted by the VGG pre-trained model [15]. Moreover, some proposed methods do not control the limitation of rotation when augmenting the training data.

As shown in Figure 1, in Odia, letters a, b, and c are three different characters. In cases where the gradient orientation in localized characters is ignored, the similarities of (a, c) and (a, b) might become closer. Hence, it is difficult to find a suitable space to represent the original data. When queries (see query 1, query 2, and query 3) arrive, it is challenging for some learning-free methods to obtain a better result by simply ranking based on similarity only. Our approach focuses on correcting the distortion and deformation of characters by considering the gradient-orientation feature and aims to find a better representation that can distinguish a character from other characters with similar structures. In Section 6, we compare the image features extracted by our pre-trained model and current existing pre-trained models, including the Vision Transformer [16].

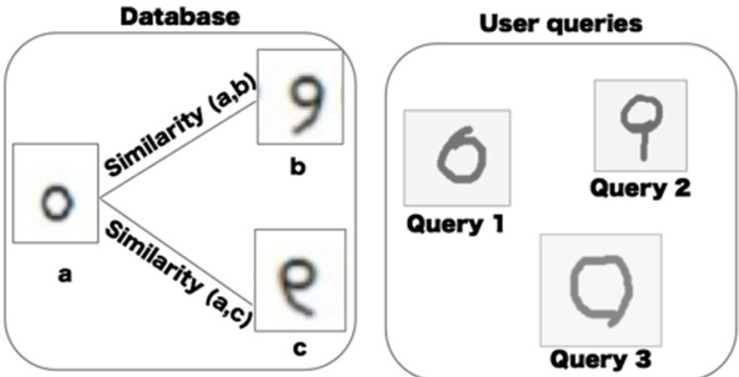

**Figure 1.** Similar characters in Odia number; a, b, and c are three different characters.

### 2.3. Meta-Learning and Metric-Based Method

Meta-learning, also known as 'learning to learn,' aims to learn models as a new method to adapt to new environments and solve several problems, such as when a classifier trained on one specific domain can tell whether a given image contains an object from another domain after seeing a handful of images. There are three common approaches to meta-learning: metric-based, model-based, and optimization-based.

Although there are many cross-domain meta-learning methods for image classification, it should be clear that the state-of-the-art will depend on the type of data. Considering the geometric characteristics and the lack of color information in character images, the methods that had results reporting character classification are selected for comparison.

For metric-learning-based methods, RelationNet [17], MatchingNet [18], and Siamese network [19] pay more attention to learning how to compare the relationship between a pair of images, which can lead to similar problems in shape-matching. It is better to learn to embed characters with similar features to the closer position in the feature space. A typical example might be prototypical networks (ProtoNet) [20], which aim to learn an embedding function to encode each input into a feature vector. Our proposed method optimizes the

performance of prototypical networks to extract geometric features and obtains better results on the benchmark datasets.

For model-based meta-learning methods, Finn et al. [21] provided a meta-learning-based method, MAML (Model-Agnostic Meta-learning), which can be applied to cross-domain character recognition and obtain good performance. To improve the cumbersome, unstable Bayesian framework, Deep Kernel Transfer (DKT) was proposed by Patacchiola et al. [22]. The Gaussian process approach is used by learning a deep kernel across tasks, and it has maintained the best record of five-way (one-shot) OMNIGLOT→EMNIST cross-domain character recognition.

Publicly available benchmark datasets and implementations of DKT and MAML provided from Patacchiola et al. [23] are used as a state-of-the-art baseline method for comparison experiments of OMNIGLOT→EMNIST cross-domain characters recognition tasks; MatchingNet [18], ProtoNet [20], and RelationNet [17] are selected as basic metric-learning-based methods to compare the performance of feature extraction; the results are introduced in Section 5.4.

## 3. Main Contributions of This Paper

The main contributions of this paper can be considered as follows:

(1) From a technical aspect, we present a new model structure based on metric learning to use a low-resource ancient character typeface dataset. It is a new attempt to apply generic handcrafted features combined with few-shot metric learning to the model, which works in low-resource data. Thus, the proposed method can obtain more low-latitude features that are conducive to the retrieval of symbols and ancient characters. Comparing with other existing metric-learning based methods, the features extracted from our proposed method have better advantages in finding the highest-ranked relevant item.

(2) Training on other sufficient public character datasets such as OMIGLOT and testing on font typeface-based datasets. This method can learn to represent the typeface images into a vector space more appropriate to their geometric properties. The method proposed in this paper also performs better than several other metric-learning based few-shot learning methods on the cross-domain task using the benchmark datasets.

(3) The calculation of the dynamic mean vector is imported to enhance the robustness of prototypical networks. In the mini-batch training process, we not only select a unique prototypical center but also adapt to the deformation of the support set from test data.

(4) Our proposed framework consists of several components, i.e., spatial transformer network, feature fusion, dynamic prototype distance calculator, and ensemble-learning-based classifier. Each component has reasonable contributions to the classification task.

(5) We apply the proposed method to retrieving ancient characters that support handwritten input, providing a new perspective for the flexible use of low-resource data. This is an innovative effort in terms of one-shot-based ancient character recognition by utilizing the metric-learning method. It provides a reference for the application of ancient font typeface resources in digital humanities and other fields in the future.

The remainder of this paper is organized as follows: In Section 4 of this paper, we describe the framework of our proposed system and present the details of the learning model, and the improvement of the proposed method is also introduced. We introduce experimental results and discussion in Section 5, and we give a summary and mention future work in Sections 6 and 7.

## 4. Methods

### 4.1. Framework Structure

Figure 2 shows the structure of our ancient-character-recognition framework.

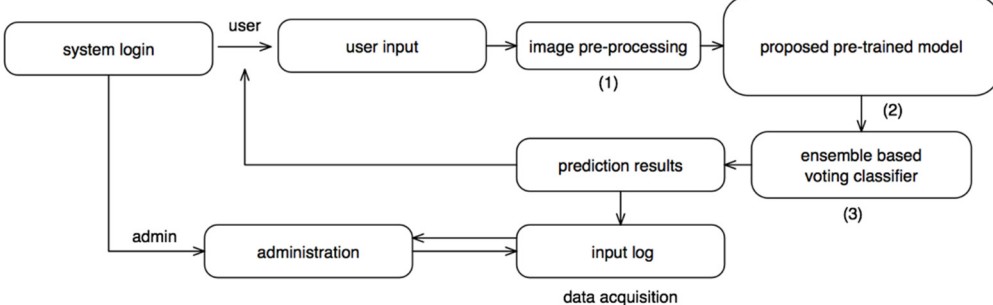

**Figure 2.** Our proposed character-recognition framework.

We describe our approaches (1)–(3) in the following sections. In Section 4.2, we discuss image pre-processing (1). In Section 4.3, we discuss the proposed pre-trained model structure (2) and voting classifier (3).

*4.2. Image Pre-Processing*

Considering the difference in character position and stroke width of typeface images, we processed the training data and user input, as shown in Figure 3.

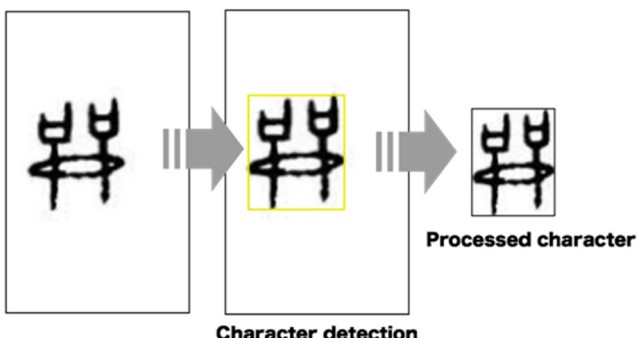

**Figure 3.** Typeface image pre-processing.

As shown on the left side of Figure 3, we used horizontal histogram projection to obtain the position of characters. The idea is to add up image columns or rows and obtain a projection whose minimum values allow us to segment characters.

*4.3. Model Structure*

The architecture of our proposed recognition model is shown in Figure 4.

It consists of four modules: (1) spatial transformer module, (2) histogram of oriented gradients (HOG) feature descriptor module, (3) feature fusion module, and (4) dynamic prototype distance calculator module. We discuss the details of our proposed modules in the later sections.

4.3.1. Spatial Transformer Module

Affine transformation helps to modify the geometric structure of an image, and it preserves collinearity and ratios of distances. It can be used in machine learning and deep learning for image processing and image augmentation [24]. It is also used to correct geometric distortions and deformations and is useful in image processing of satellite images [25]. There is a study that performed handwritten symbol classification in the presence of distortions modeled by affine transformations [26]. To solve geometric distortions and deformations problems, we utilized the spatial transformer, which is conceived in spatial transformer networks [27], in our method. The spatial transformer can reduce the influence caused by translation, scale, rotation, and more generic warping and has a better performance in handwritten character classification. It consists of the localization net and

grid generator. The localization net takes the input feature map $U_{input} \in \mathbb{R}^{H \times W \times C}$, where $H$, $W$, and $C$ represent the width, height, and channels of input images, respectively. The output of the localization net is represented by θ and calculated by $f_{loc}(U)$. The function $f_{loc}()$ can take any form, but for affine transformation, θ needs to be 6-dimensional. Hence, we designed the function $f_{loc}()$ as shown in Figure 5. From the output, we can obtain transformation parameter θ as the input of the grid generator.

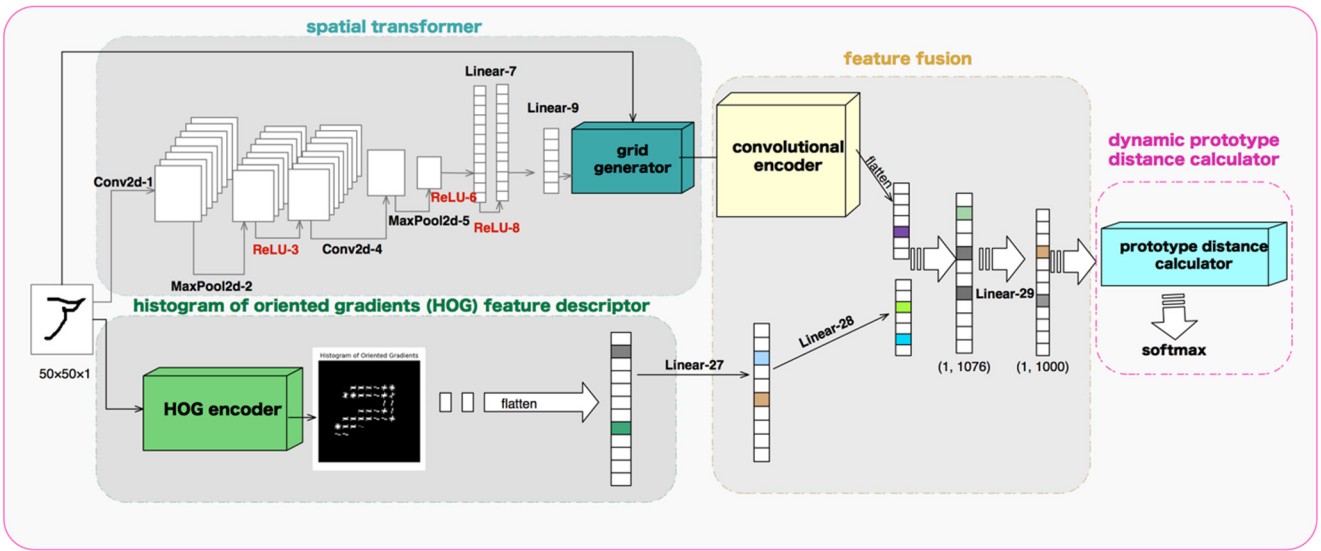

**Figure 4.** An overview of our proposed model.

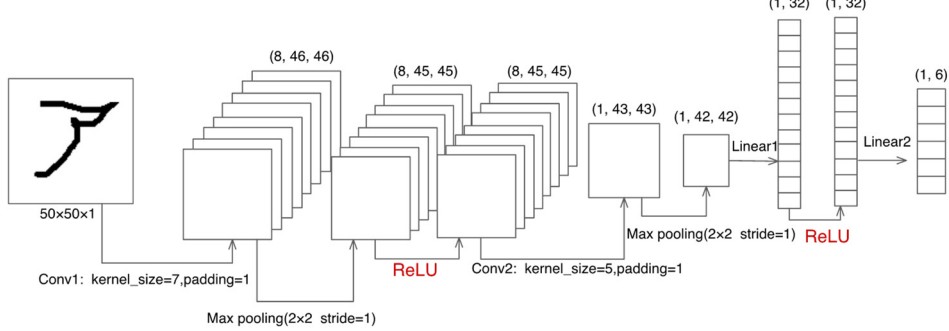

**Figure 5.** Architecture design of the function $f_{loc}()$ in our method.

The grid generator creates a sampling grid by using the calculated transformation parameters. The output of the parameterized sampling grid is represented by $V_{output} \in \mathbb{R}^{H' \times W' \times C}$, where $H'$ and $W'$ are the same as $H$ and $W$ of the input feature map. We utilized the output of spatial transformers as one part of the input of our proposed feature fusion module.

### 4.3.2. Histogram of Oriented Gradients (HOG) Feature Descriptor Module

The HOG is an efficient way to extract the gradient-orientation feature in localized portions of an image. The distribution of directions of gradients is used as features to represent an input image. The character structure has a strong orientation feature; as shown in Figure 6, ten completely different characters were randomly selected from the OMNIGLOT dataset. Though many pairs are only slightly different from each other, they can be distinguished effectively by the feature extraction of the HOG extractor.

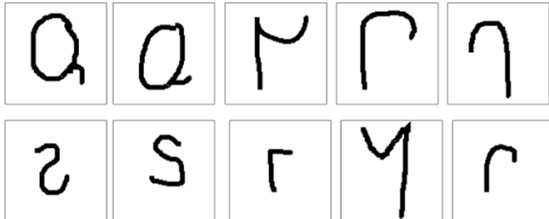

**Figure 6.** Ten completely different characters randomly selected from OMNIGLOT.

Additionally, a study [28] showed that handwritten character recognition performance could be improved by using the HOG feature as an input of a neural network. Therefore, in our work, feature extraction based on the HOG feature descriptor was applied to extract features from ancient characters.

Figure 7 shows an example of a feature map we obtained. The input images were resized to (50 × 50), we set the number of orientation bins as 8, the size of a cell as (6, 6), and number of cells in each block as (2, 2). We used a flattened HOG feature vector as the input to the 'Linear-27' calculation, as shown in Figure 4.

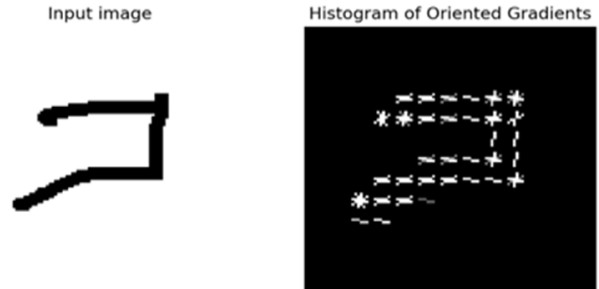

**Figure 7.** An example of HOG feature map.

### 4.3.3. Feature Fusion Module

A naive approach to combine multiple features is to concatenate the feature sets together. However, the vector may consume significant space and cannot show the combined characteristics of the object. Many studies [29,30] have learned the common representation of data from different domains by using feature-fusion technology. We used a feature-fusion module to adaptively weigh and combine the representations based on both local features extracted from the HOG feature descriptor module and features extracted from the convolutional neural network architecture. The architecture of the convolutional encoder is shown in Appendix A.

### 4.3.4. Dynamic Prototype Distance Calculator Module

As shown in Figure 8, prototypical networks [20] learn a metric space and require only a small amount of training data with limited information.

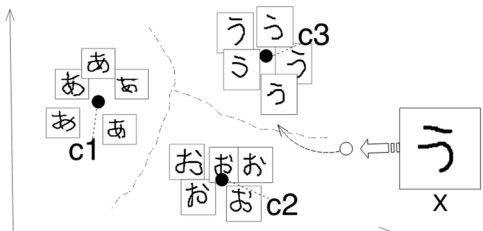

**Figure 8.** An example of prototypical networks classification in a few-shot case.

It represents each class by the mean of its examples in a representation space. The predicted probabilities of a given test input $X$ can be calculated by (1):

$$P(y = c|X) = \frac{\exp\left(-d_\varphi(f_\theta(X), V_c)\right)}{\Sigma_{c' \in C} \exp\left(-d_\varphi(f_\theta(X), V_{c'})\right)}, \tag{1}$$

where $f_\theta$ shows the embedding function, $d_\varphi$ is the distance function, and $V_c$ represents the mean vector of the embedded support data samples in this class, which is calculated by (2):

$$V_c = \frac{1}{|S_c|} \sum_{(x_i, y_i) \in S_c} f_\theta(x_i), \tag{2}$$

where $S$ is defined as a set of embedded support data samples. The loss function is defined as (3) in the training stage:

$$\mathcal{L}(\theta) = -\log P_\theta(y = c|x) \tag{3}$$

In our method, we define $d_\varphi$ as the Euclidean distance calculation and predict the input data as (4):

$$P(y = c|X') = softmax\left(-\frac{\Sigma_{i \in \{0,1,...z\}} \ d_\varphi\left(f_\theta(X'), V_{ci}\right)}{Z}\right), \tag{4}$$

where $V_{ci}$ in our training step is defined as a dynamic mean vector of a subset extracted by random sampling from the support set. The dynamic prototype distance calculator is used to calculate the loss value, and the algorithm to compute the loss $\mathcal{L}(\phi)$ for a training episode is provided in Algorithm 1.

---

**Algorithm 1** The algorithm to compute the loss $\mathcal{L}(\phi)$ in our network. Training set contains the support set S and query set Q. K is defined as the number of classes per episode. $S_q$ is the number of queries of each class in Q. RANDOMSAMPLE(S, n) denotes a set of n elements chosen uniformly at random from the set S, without replacement. $f_\theta$ is the embedding function, and $d_\varphi$ is the distance function.

---

Input:
S: support set $\{(x_{k0}, y_{k0}), (x_{k1}, y_{k1}) \ldots (x_{kn}, y_{kn}),\}$ from random selection from training set T, $k \in \{1, 2, 3, 4 \ldots K\}$, n is the number of support examples per class
Q: query set $\{(x\prime_{k0}, y\prime_{k0}), (x\prime_{k1}, y\prime_{k1}) \ldots (x\prime_{kz}, y\prime_{kz}),\}$ from random selection from training set T, there is no repetition with S, z is the number of query examples per class
Output: The loss $\mathcal{L}(\phi)$ for each training episode
Compute the length of each class in S, $Len_S = \{len_{s1}, len_{s2}, len_{s3} \ldots .len_{sk}\}$
For each class k in S generate a random integer $N_{sk} \in \{N_{s1}, N_{s2} \ldots .N_{sk}\}$ such that $0 \leq N_{sk} \leq Len_{sk}$,
For k in {1,2,3 … K} do
For $(x\prime_{ki}, y\prime_{ki})$ in $\{(x\prime_{k0}, y\prime_{k0}), (x\prime_{k1}, y\prime_{k1}) \ldots (x\prime_{kz}, y\prime_{kz}),\}$ =Q, $i \in \{0, 1, \ldots z\}$ do:
$S_{ki} \leftarrow$ RANDOMSAMPLE$(len_{sk,n})$
$c_{ki} \leftarrow \frac{1}{N_{sk}} \sum_{(x_{kj}, y_{kj}) \in S_{ki}} f_\theta\left(x_{kj}\right), \ j \in \{0, 1 \ldots N_{sk}\}$
$d_{ki} \leftarrow d_\varphi(x'_{ki}, c_{ki})$
end for

$\mathcal{L}(\phi) \leftarrow \mathcal{L}(\phi) + \frac{1}{K}\left[\frac{\Sigma_{i \in \{0,1,...z\}} \ d_{ki}}{z} \log\left(\sum_{k'} \exp\left(-\frac{\Sigma_{i \in \{0,1,...z\}} \ d_{ki}}{z}\right)\right)\right]$
end for

---

### 4.3.5. Improvement of the Classification with Ensemble Learning

As shown in Figure 9, to improve the prediction results, we used a voting classifier of ensemble methods. As there is no fine-tuning in the predicted domain, we used the K-nearest neighbors classification method based on the local distribution of the data and

the logistic regression method based on the overall distribution of the data to predict the query. We describe the result by using voting classifier combined with K-nearest neighbors and logistic regression in Section 5.2.2.

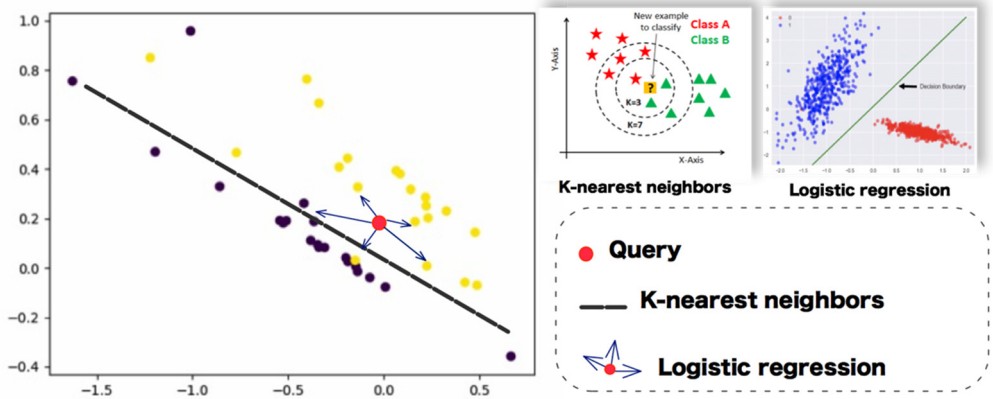

**Figure 9.** A voting classifier.

## 5. Experiments and Results

In this section, we describe our experiments. Section 5.1 introduces the databases used in the experiments and the basic experimental setup. In Section 5.2, we use the benchmark datasets on a cross-domain few-shot learning five-way task to evaluate the performance of our proposed model. Section 5.3 utilizes the proposed pre-trained model as a feature extractor to evaluate the retrieval performance compared with features obtained from other existing pre-trained models. We use the dataset described in Section 1.1 to perform this task. A comparison with the state-of-the-art methods is presented in Section 5.4. An introduction of a demo system is described in Section 5.5. The utilization of our pre-trained model in the retrieval of other historical resources is presented in Section 5.6.

### 5.1. Datasets and Basic Experimental Setup

As there is only one sample for each character in the Shirakawa font typeface dataset, a robust system framework is expected to extract features from one-shot character samples and apply these features to retrieval tasks, and there must be enough test data to evaluate the performance of the feature extraction. Hence, we evaluated our proposed method using the standard benchmark handwritten character dataset OMNIGLOT [31] for the training task, which included 1623 characters from different languages, not including oracle bone characters, and 20 samples for each character. For the test task, we used EMNIST [32], which included 47 characters based on a mix of letters and digits and 2400 samples for each character. We used these two datasets on a regular five-way (five-shot), cross-domain, few-shot learning task. We used model weights trained on OMNIGLOT to extract the features from the Shirakawa oracle bone font typeface [2] to evaluate the performance of the proposed model. We manually collected 40 queries from the website [33], used an automatic segmentation algorithm to extract all oracle bone characters on the OMNIGLOT webpage, and labeled all the segmented characters according to the modern character annotations of the OMNIGLOT webpage. We performed feature extraction and retrieval experiments on 681 oracle bone characters using the pre-trained model. To show the scalability in different font retrievals, two font families, MERO_HIE hieroglyphics font [3] and Aboriginebats font [4], were used to display the vector visualization and the example of the retrieval results. We used Adamax [34] as the optimizer, and the learning rate was set to 0.001.

### 5.2. Cross-Domain, Few-Shot Classification Performance

5.2.1. Cross-Domain, Few-Shot Classification Performance on the 'OMNIGLOT→EMNIST' Task

To test the ability of feature extraction learned by the system on unseen input data, we evaluated the proposed method on the 'OMNIGLOT→EMNIST' task, which means training on OMNIGLOT, extracting features on EMNIST for classification testing, and not performing fine-tuning on EMNIST. Each epoch randomly selects five categories of data from OMIGLOT, and each category includes 1 support sample and 15 queries. Figure 10 shows the losses on the training domain and the target domain under different batch size settings. The results showed that a model with higher accuracy in the target domain can be obtained under a smaller batch size setting. However, in this case, the classification performance of the trained model in the target domain is unstable.

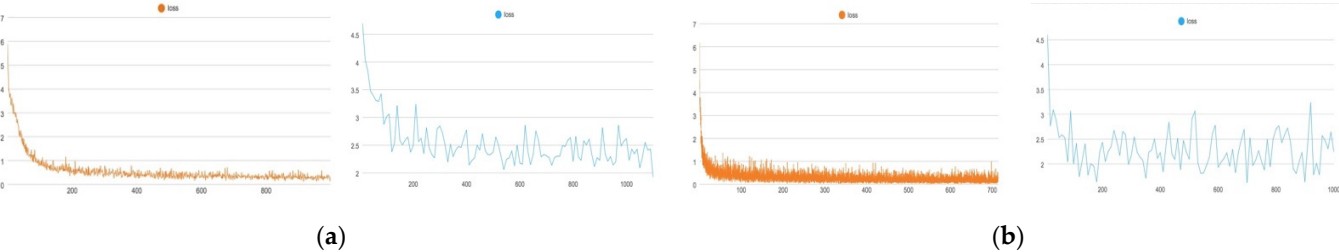

| **(a)** | **(b)** |

**Figure 10.** (**a**) Batch size 100, (**b**) Batch size 50. The loss value in training domain and target domain at 1000 epochs. The orange line represents the training domain, the blue line represents the target domain, the vertical axis represents the loss value, and the horizontal axis represents the number of epochs.

Table 2 shows the comparative experimental results of the five-way (one-shot) classification task between the primary metric learning baseline methods and our method. The results showed that our proposed method has learned a better representation from characters than ProtoNet [20] in the five-way (one-shot) classification task. It has a good performance in the best sampling situation, but compared to MatchingNet [18] and RelationNet [17], the average classification accuracy of the five-way task is lower.

**Table 2.** Comparison on five-way (one-shot) classification task.

| Method | Average |
|---|---|
| MatchingNet (Vinyals et al. 2016) [18] | $75.01 \pm 2.09$ |
| ProtoNet (Snell et al. 2017) [20] | $72.77 \pm 0.24$ |
| RelationNet (Sung et al. 2018) [17] | $75.62 \pm 2.00$ |
| Our proposed method | $74.00 \pm 6.00$ |

5.2.2. Evaluation of the Ensemble-Learning-Based Classification Method

To evaluate the performance of the classifier used on the target domain and to compare the performance of the ensemble-learning-based classification method proposed in Section 4.3.5, we tested the classification ability of each classifier under different settings of sample numbers in the support set and different feature dimensions. The results are shown in Table 3.

The experimental results indicated that logistic regression has a good classification ability in the five-shot and ten-shot classification tasks with more samples in the support set. Compared to the one-shot classification task with fewer samples in the support set, using low-dimensional features in 50 dimensions, K-nearest neighbors has robust distance calculation and classification capabilities. Still, our proposed ensemble learning method has a better performance in the case of increased feature dimensions and has the best performance in the one-shot task.

**Table 3.** Classification ability comparisons on each classifier under different settings of sample numbers in the support set and different feature dimensions.

| Methods | Number of Samples in Support Set | Average Score | | |
|---|---|---|---|---|
| | | 50 Dimensions | 500 Dimensions | 1000 Dimensions |
| K-nearest neighbors | one shot | 0.74 | 0.5 | 0.66 |
| | five shots | 0.78 | 0.48 | 0.8 |
| | ten shots | 0.80 | 0.84 | 0.82 |
| Logistic regression | one shot | 0.64 | 0.48 | 0.68 |
| | five shots | 0.78 | 0.74 | 0.92 |
| | ten shots | 0.94 | 0.72 | 0.90 |
| Ensemble learning (proposed method) | one shot | 0.68 | 0.64 | 0.74 |
| | five shots | 0.76 | 0.72 | 0.83 |
| | ten shots | 0.88 | 0.82 | 0.86 |

### 5.2.3. Performance on the Forty-Way Classification Task

Our test target domain EMNIST has more than 40 characters; thus, we set the number of categories trained on OMNIGLOT to 40. We tried the forty-way (five-shot) training task; 40 categories of characters were sampled for training on the training domain for each epoch, and each category had 15 queries for training.

Table 4 shows the performance of the features obtained from the trained model utilized in the classification task with different category settings of the target domain. For each category in the target domain, we used only one-shot support data; thus, we regarded it as a retrieval task and used precision at 1, 5, and 10 (P@1, P@5, P@10) to evaluate the performance.

**Table 4.** The performance of the retrieval task that uses the features extracted from the model trained on forty-way (5-shot) task and 1000 epochs.

| Number of Classification Categories | P@1 | P@5 | P@10 |
|---|---|---|---|
| 20 | 0.55 | 0.85 | 0.9 |
| 30 | 0.3 | 0.83 | 0.9 |
| 47 * | 0.28 | 0.65 | 0.76 |

* On the test domain, we used all the 47 categories of data from EMNIST.

It can be seen from the results that the accuracy of the top one decreases after expanding the number of classification categories, but the obtained features are still suitable for retrieval tasks that show 10 results as the results in P@10.

### 5.2.4. Ablation Experiments

As the expected target ancient character font typefaces [2–4] usually include more than 40 categories, we conducted ablation experiments of our proposed model on the forty-way training task.

Skeletonization was used as one of the comparison targets of the ablation experiments. Skeletonization is a process of reducing foreground regions in a binary image to a skeletal remnant that largely preserves the extent and connectivity of the original region while throwing away most of the original foreground pixels. Skeleton maps, as shown in Figure 11, provide an efficient shape descriptor in many applications, such as content-based image-retrieval systems and character-recognition systems. We used the skeletonization method of Zhang et al. [35] and set a comparative experiment with the original image for training.

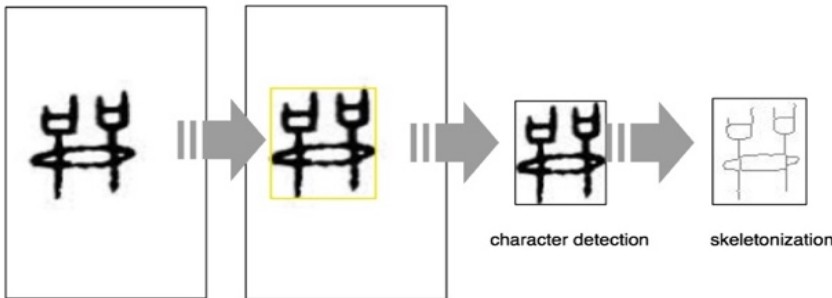

**Figure 11.** Skeletonization process.

It can be seen from Table 5 that the spatial transformer plays a significant role in our model. If the model does not use the dynamic prototype distance calculator proposed in Section 4.3.4., the accuracy will be affected, even lower than if the HOG encoder module is not used. As shown in the results, for current data, the skeletonization method does not improve the results.

**Table 5.** Ablation experiments of our proposed model.

| No. | HOG Encoder [1] | Spatial Transformer [2] | Dynamic [3] | Skeletonization [4] | P@1 | P@5 | P@10 |
|---|---|---|---|---|---|---|---|
| 1 | ✔ | ✖ | ✔ | ✖ | 0.19 | 0.43 | 0.63 |
| 2 | ✖ | ✔ | ✔ | ✖ | 0.24 | 0.61 | 0.73 |
| 3 | ✔ | ✔ | ✖ | ✖ | 0.17 | 0.52 | 0.6 |
| 4 | ✔ | ✔ | ✔ | ✖ | 0.28 | 0.65 | 0.76 |
| 5 | ✔ | ✔ | ✔ | ✔ | 0.20 | 0.26 | 0.41 |

[1] HOG feature descriptor module described in Section 4.3.3.; [2] spatial transformer module described in Section 4.3.1.; [3] dynamic prototype distance calculator module described in Section 4.3.4.; [4] skeletonization process mentioned in this section.

The visualization of the training process introduced in this section can be found in Appendix B, Figure A1, and Table A2.

## 5.3. Contrastive Experiment of Features Extracted from Pre-Trained Models for the Retrieval Task

In actual use, there is only one image for each character of each font resource. We compared the performance of the features extracted from the existing pre-training models and our proposed model in the retrieval task.

We prepared 40 oracle bone characters obtained from the OMNIGLOT website [33] that eliminated noncorresponding data, which means characters completely different from those recorded in the Shirakawa font. Examples of noncorresponding data are shown in Appendix C and Table A3.

The search range is 681 oracle bone characters from the Shirakawa font [2]. We chose the VGG19 [36] and ResNet50 [37] models that are most commonly used in image-retrieval studies and the transformer-based model ViT [16] for comparison. As we used a total of 681 oracle bone characters from the Shirakawa font, the index for retrieval was increased to 681.

We used the evaluation metric MRR (Mean Reciprocal Rank) to evaluate the system. Each query has only one corresponding correct result. We set the experiments using different models to extract features and applied them to retrieval tasks to prove the performance of our method of feature extraction.

Table 6 shows that as our model is not trained on a dataset such as ImageNet, the results shown in Table 6 indicate that the features extracted by our model are more suitable for use in character retrieval without color information. It can be seen from the results that our system has a better performance in finding the highest-ranked relevant item. The source code and queries used for evaluation in this section are explained in Section 7.

**Table 6.** Performance evaluation of pre-training features on retrieval tasks.

| Model | Feature Dimension | MRR Score |
| --- | --- | --- |
| VGG19 [36] | 1000 | 0.0580 |
| ResNet50 [37] | 1000 | 0.0563 |
| ViT [16] | 1000 | 0.0586 |
| Our proposed method | 1000 | 0.1943 |

Our method can also be generalized to other font datasets. Table 7 shows the vector visualization of our pre-trained model applied in the MERO_HIE hieroglyphics font and Aboriginebats font datasets [3,4].

**Table 7.** Application performance on MERO_HIE hieroglyphics font and Aboriginebats font datasets.

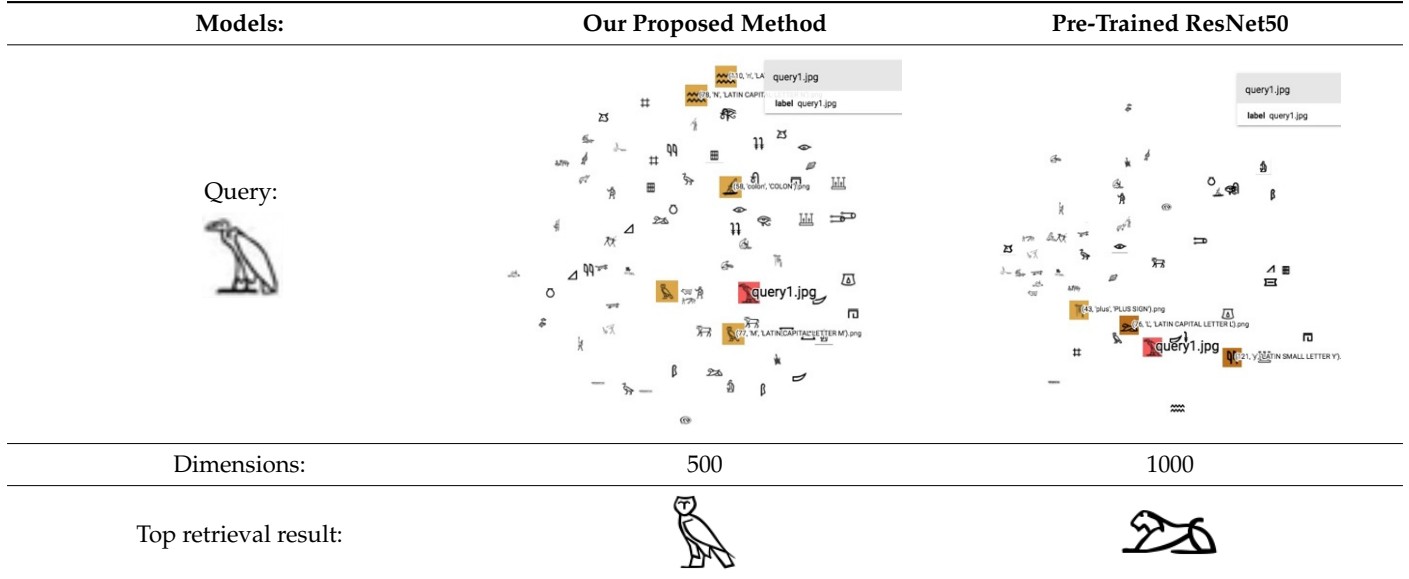

| Models: | Our Proposed Method | Pre-Trained ResNet50 |
| --- | --- | --- |
| Query: | | |
| Dimensions: | 500 | 1000 |
| Top retrieval result: | | |

A simple stroke of a bird was used as a query. It can be seen that the features extracted by our model are more sensitive in shape-matching, and better results can be obtained. In the visualization figure, the red box shows the most similar image based on the cosine similarity calculation, and the yellow box represents the top five search results.

### 5.4. Comparison with the State-of-the-Art Methods

As the data used in our utilization have only one sample per category, comparison with state-of-the-art methods is based on the same 'OMNIGLOT→EMNIST' task. The current state-of-the-art methods based on meta-learning in the same 'OMNIGLOT→EMNIST' task were selected as the DKT [22] and MAML [21] to conduct a comparative experiment.

As there is a difference in the total number of categories of the EMNIST dataset used in Section 5.2 compared to the DKT experimental dataset setting [23] (Patacchiola et al., 2020), we used the implementation contribution and datasets provided by Patacchiola et al. [22] to conduct comparative experiments.

The results on five-way one-shot and five-shot tasks are shown in Table 8. Compared with the test data we used in Section 5.2, the results are improved. On the one-shot task, the result from our proposed method is unstable but can obtain a better maximum accuracy. The DKT proposed by Patacchiola et al. has a good performance on five-way classification tasks.

**Table 8.** Results for OMNIGLOT→EMNIST one-shot classification.

| Method | One-Shot | | Five-Shot | |
|---|---|---|---|---|
| | **Reported** | **Our Re-Tested Results** | **Reported** | **Our Re-Tested Results** |
| MAML (Finn et al., 2018) | 72.04 ± 0.83 | 71.12 ± 0.84 | 88.24 ± 0.56 | 88.80 ± 0.24 |
| DKT + BNCosSim (Patacchiola et al., 2020) | 75.40 ± 1.10 | 74.90 ± 0.71 | 90.30 ± 0.49 | 90.11 ± 0.20 |
| DKT + CosSim (Patacchiola et al., 2020) | 73.06 ± 2.36 | 76.00 ± 0.42 | 88.10 ± 0.78 | 89.31 ± 0.19 |
| Our proposed method | (-) | 73.33± 2.67 | (-) | 83.31 ± 0.87 |

MAML and DKT do not support feature extraction for retrieval. Metric-learning-based methods MatchingNet, ProtoNet, and RelationNet were used for comparison as feature extractors. We retrained these three architectures, and the 'Conv4' mentioned in [22] was used as the backbone network training and feature extraction. Under the evaluation data and methods introduced in Section 5.3, the results shown in Table 9 were obtained. From the results, it can be seen that the features extracted from our proposed method have better advantages in finding the highest-ranked relevant item.

**Table 9.** Performance evaluation of pre-training features on retrieval tasks.

| Model | Feature Dimension | MRR Score |
|---|---|---|
| RelationNet (Sung et al., 2018) | 1600 | 0.1022 |
| MatchingNet (Vinyals et al., 2016) | 1600 | 0.0605 |
| ProtoNet (Snell et al., 2017) | 1600 | 0.0817 |
| Our proposed method | 1000 | 0.1943 |

*5.5. Demo Application Implementation*

A demo application was implemented for character recognition based on our proposed method. Figure 12 shows the user interface of the application implementation.

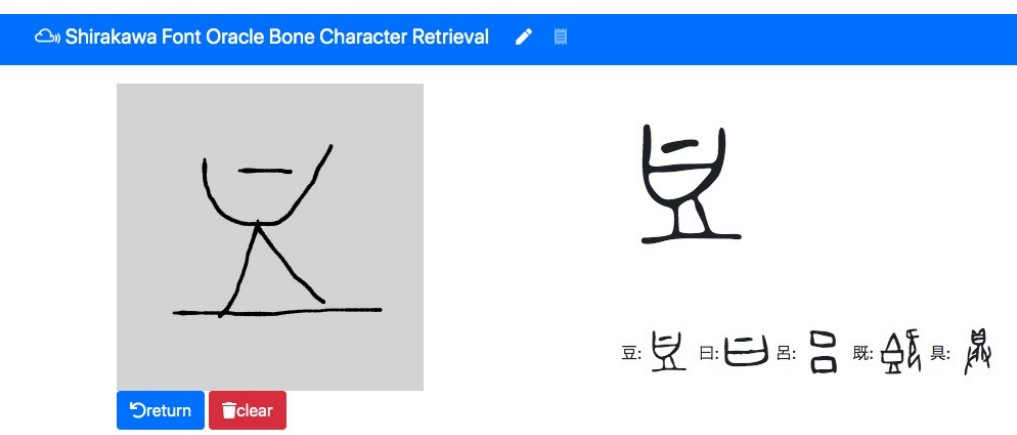

**Figure 12.** Demo application implementation.

Each stroke sketched by a user is recorded by the system, and the search results are updated when the user completes the input of a stroke. Each user input stroke is converted into an image for a prediction. Each prediction does not rely on trajectory information of the user input. As most of the oracle bone characters correspond to a modern kanji character, the retrieval results are shown as pairs of ancient characters and their corresponding modern characters. Ancient characters are not represented by images, but they are represented directly by the Shirakawa font, which is explained in Section 1.1.

*5.6. Other Utilization*

In addition to the application described in Section 5.5, our method can also be applied to unlabeled valuable historical documents. The segmentation method is the same as the

character-detection method used in Section 4.2. Figure 13 shows a case of utilization of the proposed method for sorting out unlabeled manuscripts [38] of the oracle bone script. The features extracted from our model were used for unlabeled image clustering, and the visualization of the actual results is shown in Figure 13.

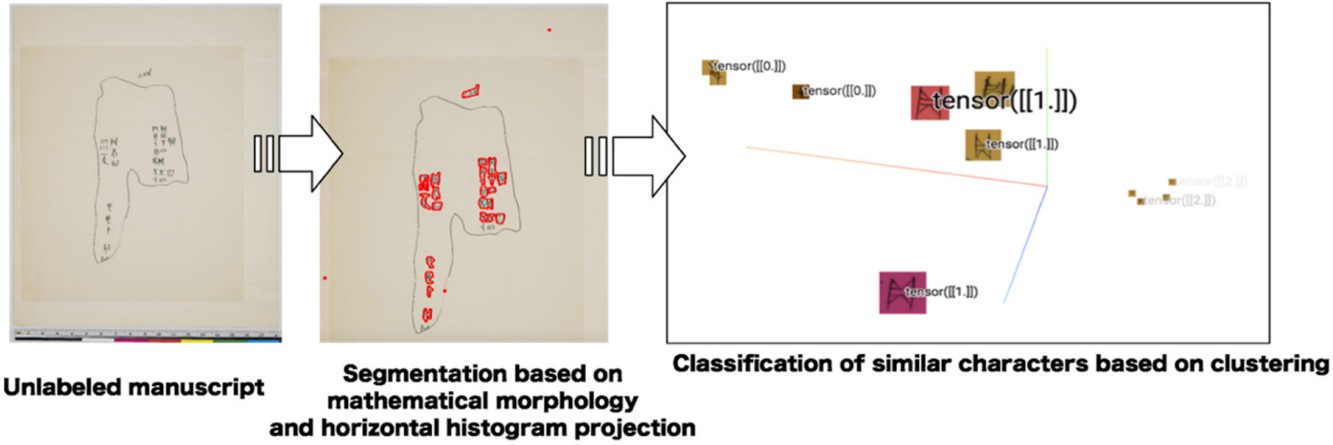

**Figure 13.** Utilization of the proposed method for recognition of oracle bone characters from historical documents.

## 6. Discussion

From the results of experiments and analysis, it can be seen that the existing deep-learning models trained on large-scale data have limitations in extracting features from images with solid geometric characteristics. In our previous work, we conducted a comparative experiment [39] on the features of different layers extracted from the VGG19 pre-trained model. The goal of this study was to utilize the one-shot font typeface images to achieve a good retrieval performance. We found that, unlike the retrieval of artwork images [15], typeface images with weaker color and texture features have some issues that need to be solved, such as dealing with geometric deformation and detailed geometric features.

Comparative details are given in Table 10. Compared with the existing approaches, our work that uses only one reference sample for each category to retrieve more than 500 characters is a novel attempt. However, solving the problem of different character variation problems mentioned in Appendix C is a challenge for future studies and attempts.

**Table 10.** Comparison between this research and other exiting methods in different tasks.

| Task | Our Proposed Method | Model-Based Meta-Learning Methods | Metric-Learning Based Methods | CNN Based and Transformer Based Pre-Trained Models |
|---|---|---|---|---|
| Five-way classification tasks on benchmark dataset | The performance is not as good as the model-based meta-learning method, which is relatively unstable, but has relatively good best results. | Bayesian framework Deep Kernel Transfer (DKT) proposed by Patacchiola et al. [22] has the best results but consumes more training resources. | RelationNet (Sung et al., 2018) has good results, but it is not as good as the model-based meta-learning method on the cross-domain character images classification task. | It is not a common method of few-shot learning and has not been evaluated by this research. |

**Table 10.** *Cont.*

| Task | Our Proposed Method | Model-Based Meta-Learning Methods | Metric-Learning Based Methods | CNN Based and Transformer Based Pre-Trained Models |
|------|---------------------|-----------------------------------|-------------------------------|----------------------------------------------------|
| Used as feature-extractor for character images | Due to the focus on geometric feature processing and feature extraction, our model has a better performance on character data when used as a feature-extractor. | Due to insufficient descriptions and cases for feature extraction, this research did not conduct evaluation experiments using such methods. | Due to the use of pair images for learning, methods such as RelationNet and MatchingNet are more suitable for use as feature-extractor for the identification of the authenticity of handwritten characters. ProtoNet (Snell et al., 2017) is more suitable for use as a feature-extractor for retrieval, but if it is character data, some optimizations that focus on the use of geometric features are worth recommending. | Used large-scale datasets such as ImageNet for training, which is very effective for real-life image feature extraction with texture and color features. However, when extracting character data with obvious geometric features, performance needs to be improved. |

## 7. Conclusions

In this work, we proposed a feature fusion spatial transformer structure combined with a prototypical network, which focuses on achieving font-typeface-based ancient character script handwriting recognition and correcting the distortion and deformation of handwritten characters to improve the accuracy of recognition in low-resource datasets.

Our proposed method could be useful to researchers looking for new perspectives for taking advantage of scarce ancient characters or symbol typeface records.

Figure 14 shows that our method can also be applied to approximate query matching for retrieval-based character recognition, and it is very convenient to combine with other model architectures. In future work, we will focus on reducing the unstable problems of our proposed model. Additionally, we will aim to improve the generalization of the features extracted by our model while improving the prediction accuracy of the classifier.

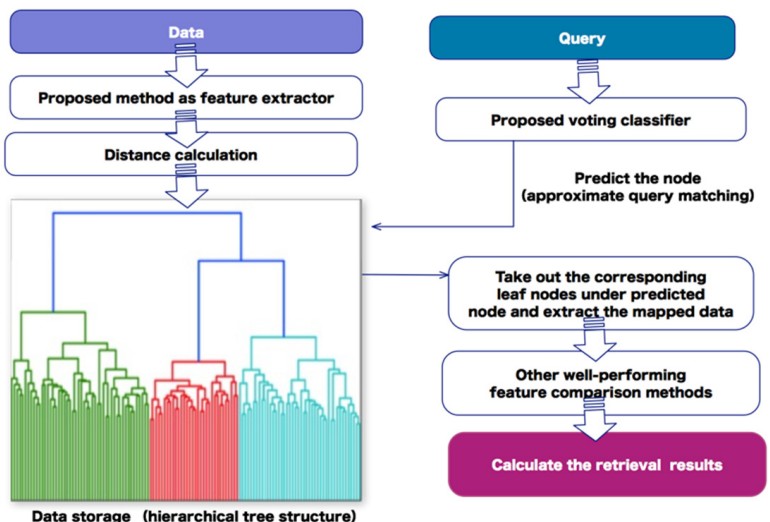

**Figure 14.** Our proposed method applied to approximate query matching for retrieval-based character recognition.

**Author Contributions:** Conceptualization, K.L., B.B. and A.M.; Methodology, K.L.; Resources, B.B. and A.M.; Data curation, B.B.; Software, K.L.; Funding acquisition, A.M. and K.L.; Project administration, A.M.; Writing—original draft, K.L.; Writing—review & editing, B.B. and A.M. All authors have read and agreed to the published version of the manuscript.

**Funding:** This research was funded by JSPS KAKENHI Grant Numbers JP 20K12567 and JP 21J15425.

**Institutional Review Board Statement:** Not applicable.

**Informed Consent Statement:** Not applicable.

**Data Availability Statement:** The Shirakawa oracle bone font: The Shirakawa Shizuka Institute of East Asian Characters and Culture, Shirakawa font project. Available online: http://www.ritsumei.ac.jp/acd/re/k-rsc/sio/shirakawa/index.html, accessed on 16 November 2021. The algorithm implementation is available on GitHub at https://github.com/timcanby/FFSTP-Feature-Fusion-Spatial-Transformer-Prototypical-Networks, accessed on 16 November 2021. The demo system is available online: http://kkocr.net/, accessed on 16 November 2021.

**Acknowledgments:** This work was partially supported by JSPS KAKENHI, grant numbers JP 20K12567 and JP 21J15425.

**Conflicts of Interest:** The authors declare no conflict of interest.

## Appendix A

The architecture of the convolutional encoder mentioned in Section 4.3.3 and Figure 4 is shown in Table A1.

**Table A1.** Architecture of the convolutional encoder.

| Layer | Settings |
| --- | --- |
| Conv2d-10 | kernel_size = (3, 3), stride = (1, 1), padding = (1, 1) |
| Conv2d-11 | kernel_size = (3, 3), stride = (1, 1), padding = (1, 1) |
| ReLU-13 | (-) |
| MaxPool2d-14 | kernel_size = 2, stride = 2, padding = 0, dilation = 1 |
| Conv2d-15 | kernel_size = (3, 3), stride = (1, 1), padding = (1, 1) |
| BatchNorm2d-16 | eps = $1 \times 10^{-5}$, momentum = 0.1 |
| ReLU-17 | (-) |
| MaxPool2d-18 | kernel_size = 2, stride = 2, padding = 0, dilation = 1 |
| Conv2d-19 | kernel_size = (3, 3), stride = (1, 1), padding = (1, 1) |
| BatchNorm2d-20 | eps = $1 \times 10^{-5}$, momentum = 0.1 |
| ReLU-21 | (-) |
| MaxPool2d-22 | kernel_size = 2, stride = 2, padding = 0, dilation = 1 |
| Conv2d-23 | kernel_size = (3, 3), stride = (1, 1), padding = (1, 1) |
| BatchNorm2d-24 | eps = $1 \times 10^{-5}$, momentum = 0.1 |
| ReLU-25 | (-) |
| MaxPool2d-26 | kernel_size = 2, stride = 2, padding = 0, dilation = 1 |

## Appendix B

As shown in Figure A1, we selected three classes of characters from OMNIGLOT, which are not related to the test domain in the structure of graphics, as the training domain.

We set the output size of the 'Linear-29' layer as two and used two classes of other characters for testing.

The training process is shown in Table A2. The scatter points of different colors in the figure represent the coordinate information output by different classes of characters from the 'Linear-29' layer of our model.

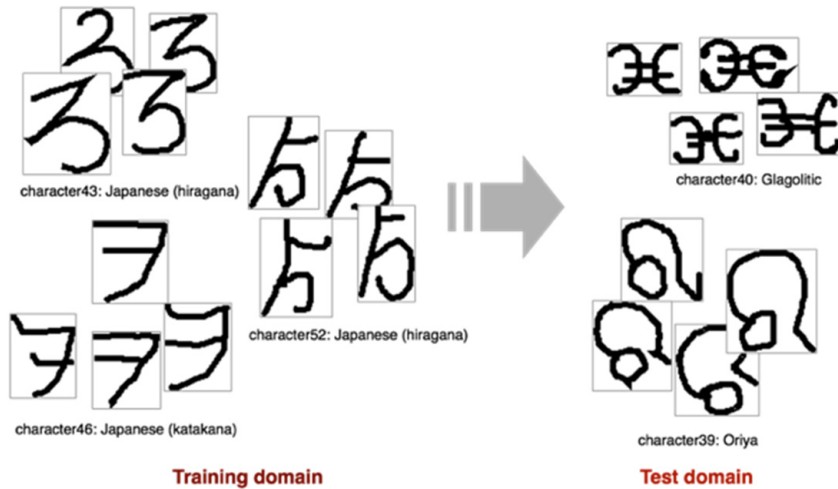

**Figure A1.** Training domain and test domain.

**Table A2.** Visualization of training processes.

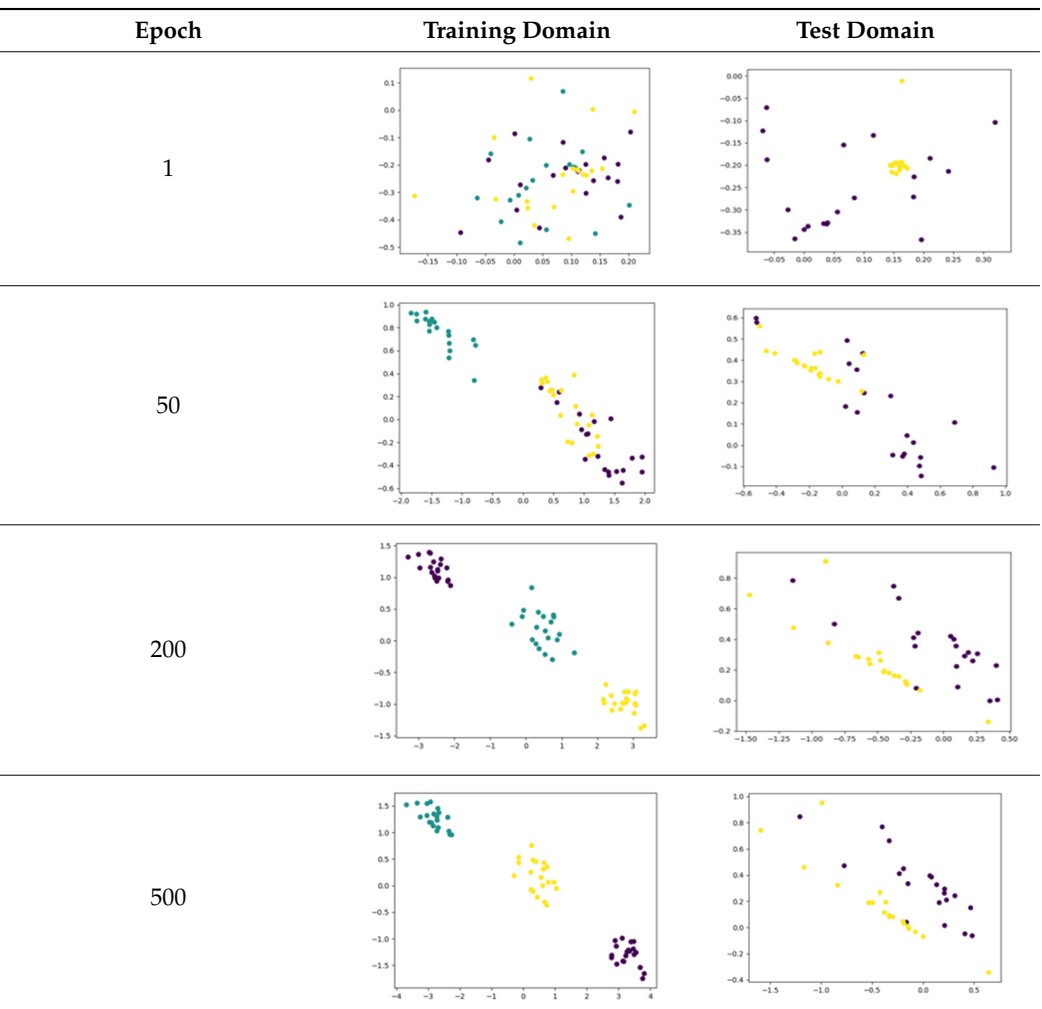

As can be seen from Table A2, although we did not fine-tune our model on the test domain, the output of the 'Linear-29' layer of our model in the test domain is gradually becoming linearly separated. It becomes adapted to the voting classifier in our proposed framework.

**Appendix C**

Queries removed from the test data are shown in Table A3. As shown in the table, there is a significant difference between the query from OMNIGLOT and our font data from the Shirakawa font. This may be caused by different variations in the same character; hence, in our experiment, we did not use all 21 characters. Table A3 shows only four examples.

**Table A3.** Examples of queries removed from the test data.

| Label Japanese (English) | Shirakawa Font [2] | OMNIGLOT [33] |
|:---:|:---:|:---:|
| 魚 (fish) |  |  |
| 馬 (horse) |  |  |
| 妻 (wife) |  |  |
| 光 (light) |  |  |

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
