# Peer review of "A Prototypical Network-Based Approach for Low-Resource Font Typeface Feature Extraction and Utilization"

_data_

Round 1
Reviewer 1 Report
The paper introduces a framework for retrieving low resource font typeface databases by handwritten input. A new deep learning model structure based on metric learning is proposed to extract the features of character typeface and used for predicting the category of handwriting input queries.
The paper starts with a quite brief introduction where subsection 1.1 is missing. Then it continues with section 2 on related work, that includes a review of ancient character recognition in subsection 2.1, a review of image retrieval based on shape matching in subsection 2.2, a paragraph on Meta-learning and metric-based method in subsection 2.3, and a brief analysis of the main contribution of this paper in subsection 2.4. Section 3 follows, with a description of the researchs method. The paper continues with Section 4, on experiments and results, followed by a one-paragraph Section 5 on discussions, and a one-paragraph Section 6 on conclusions. Three appendices end the paper.
The structure of the paper is quite strange. Also, the lack of detail of more sections should be remedied. I advise the following:
- Correct the issue of the missing subsection 1.1.
- Expand the subsection 2.3 on meta-learning and metric-based method.
- Move the subsection with the main contributions of the paper out of the section with related work.
- Section 4 should include a comparison with state-of-the-art methods.
- Expand sections 5 and 6 with discussion and conclusions in order to be relevant in the economy of the paper.
- Revise the style of English, if possible with the help of a native speaker.
Author Response
Thank you for allowing us to submit a revised draft of the manuscript “A Prototypical Network-Based Approach for Low-Resource Font Typeface Feature Extraction and Utilization”.
Below, we have attached our response outlining the explanations that will accompany the revised vision of our paper.
We really appreciate the time and effort you dedicated to providing feedback on our manuscript and are grateful for the insightful comments to our paper. We would like to express our gratitude for your sincere guidance in advising us with revising and structuring our paper.

Reviewer 2 Report
Overall a valuable article from a scientific point of view. Backed by an appropriate literature review. The methodology of the conducted experimental research is very well described.
Critical comments to the article:
- The article shoudl be written in the passive voice.
- Figure 9 is not legible.
Author Response
Thank you for allowing us to submit a revised draft of the manuscript “A Prototypical Network-Based Approach for Low-Resource Font Typeface Feature Extraction and Utilization”.
Below, we have attached our response letter outlining the explanations that will accompany the revised vision of our paper. We appreciate the time and effort you dedicated to providing feedback on our manuscript. As suggested, within the revised version of our paper, we have polished English language writing by correcting grammatical errors, reconstructing sentences, and modifying some parts of the article by using the passive voice. We would like to express our gratitude for your sincere guidance in advising us with expanding and revising our paper.

Reviewer 3 Report
The paper entitled " A prototypical network-based approach for low resource font typeface feature extraction and utilization" describes the feature fusion spatial transformer structure combined with prototypical network, which focuses on achieving a font typeface-based ancient character script handwriting recognition and correcting the distortion and deformation of handwritten characters to improve the accuracy of recognition on low resource datasets
The topic of the paper is interesting and important. The described research, as well as used methods, justify a choice of the journal. The introduction provides sufficient background. The work is properly designed and performed. The presented data are original. The methodologies of data collection are described in detail. In the case of data quality, all requirements were met.
However, I have some suggestions.
- The numbering of the references in the main text should be corrected.
- The conclusion should be completed and strengthened
- Please, revise carefully English.
Author Response
Thank you for allowing us to submit a revised draft of the manuscript “A Prototypical Network-Based Approach for Low-Resource Font Typeface Feature Extraction and Utilization”.
Below, we have attached our response outlining the explanations that will accompany the revised vision of our paper.
We really appreciate the time and effort you dedicated to providing feedback on our manuscript and are grateful for the insightful comments and valuable improvements to our paper. We would like to express our gratitude for your sincere guidance in advising us with revising and structuring our paper.
